# A universal method for depositing patterned materials in situ

Yifan Chen[1,5], Siu Fai Hung[1,5], Wing Ki Lo [1], Yang Chen[1], Yang Shen[1], Kim Kafenda [1], Jia Su[2,3], Kangwei Xia [1✉] & Sen Yang [1,4✉]

Current techniques of patterned material deposition require separate steps for patterning and material deposition. The complexity and harsh working conditions post serious limitations for fabrication. Here, we introduce a single-step and easy-to-adapt method that can deposit materials in-situ. Its methodology is based on the semiconductor nanoparticle assisted photon-induced chemical reduction and optical trapping. This universal mechanism can be used for depositing a large selection of materials including metals, insulators and magnets, with quality on par with current technologies. Patterning with several materials together with optical-diffraction-limited resolution and accuracy can be achieved from macroscopic to microscopic scale. Furthermore, the setup is naturally compatible with optical microscopy based measurements, thus sample characterisation and material deposition can be realised in-situ. Various devices fabricated with this method in 2D or 3D show it is ready for deployment in practical applications. This method will provide a distinct tool in material technology.

[1] Department of Physics, The Chinese University of Hong Kong, Shatin, New Territories, Hong Kong, China. [2] Department of Biology, South University of Science and Technology of China, Shenzhen, Guangdong 518058, China. [3] Shenzhen 34 Technology Co., Ltd, Qianhai, China. [4] Shenzhen Research Institute, The Chinese University of Hong Kong, Shatin, New Territories, Hong Kong, China. [5]These authors contributed equally: Yifan Chen, Siu Fai Hung. ✉email: kangweixia@gmail.com; syang@cuhk.edu.hk

Conventional manufacturing processes, ranging from circuit board printing down to integrated circuit and nano-device fabrication, consist of multiple steps such as mask production, material deposition, photolithography, and lift-off. Not only does each step add cost and chance to fail, it puts demanding requirements. For example, the sample surface temperature can be heated up too high during electron beam evaporation. Organic chemicals used in the photolithography process or high vacuum in deposition process can degrade the sample quality and put further limitations on the sample to be used. Especially during lift-off, faults can lead to a waste of efforts and materials. And owing to the complication of these procedures, advanced equipment and intensive training are required. From the application perspective, precise positioning is a common difficulty under the current protocol. Owing to the multi-step nature of photolithography, samples covered by photoresist have to be moved among different setups during each procedure. Imaging through photoresist and the re-alignment after each movement can deteriorate the fidelity. Targeting on special sample locations for fabrication while simultaneously evaluating the sample quality is also difficult, because most fabrication processes are not compatible with most characterisation measurements.

To invent a general patterned material deposition method with single step is desired to solve these problems. One of the key challenges is to find a mechanism to pattern the materials while depositing them in situ. The mechanism has to be general enough that it can be applied to commonly used materials, and the deposition should be large and thick, and have a quality suitable for real applications. There are a variety of ways for patterning polymer, including thermal process, photochemistry and optical trapping[1–3]. However, to pattern large scale inorganic materials directly, especially metals is still challenging. Up to now, only photon-based methods can reach both high resolution and accuracy. Different methods are based on different light induced effects. The methods based on photothermal induced physical form changes usually require high power lasers, nevertheless fabrication with high resolution and with materials like noble metals in small scales are difficult. Laser-induced chemical reactions, such as polymerisation and reduction, can deposit nanostructures with limited selected materials[4–6]. Optical trapping can also be used for depositing nanostructures either directly from their solution[7–9], or from photon-induced chemical reactions[10] However, without a bonding mechanism among particles, both the mechanical strength and electrical conductance can be compromised in the finished structures. There are also approaches to build metal composite structures by mixing metal particles with polymer and using polymer printing techniques. But the conductivity of these composite structures is usually orders of magnitude smaller than the bulk value. Also, for some of the polymer-based methods, after the deposition, the development of samples with organic solvents is required, this posts limitations on both materials to be deposited and the substrate to be used. To solve all these problems, in this work, we present a single-step direct material deposition method based on the combination of the photon-induced chemical reduction and laser-induced optical trapping processes via semiconductor nanoparticles as the agent. The nanoparticles not only become a universal reduction agent and broaden the selection of materials, but also can be work as the seeds of growth trapped by the optical force and can act as a 'glue' to form rigid composite depositions. This approach enables us to deposit a broad selection of materials with large area and thickness, high quality and high accuracy. It removes fabrication limitations posted by traditional methods and makes devices ready for practical applications.

The working principle of the light-induced material deposition (LIMD) is illustrated in the schematic drawings in Fig. 1a. The reagent of this LIMD method consists of two water based solutions: Part A, contains mainly metallate; Part B, contains semiconductor nanoparticles. When a continuous wave (CW) laser is focused on the reagent, free electrons, excited from the valence band of the semiconductor nanoparticle by photons, trigger a chemical reduction process, which converts metal ions in the solution to metal particles on the surface of the semiconductor nanoparticle. Simultaneously, the focused laser beam also works as an optical trap driving particles towards the focus spot near the substrate surface[11,12]. Unlike commonly used water-solvable reduction agents, these semiconductor nanoparticles can couple these two processes together. Thus, the chemically-reduced metal growing on the surface of the particles works like a glue, bonding trapped particles together to form a mechanically rigid metal/semiconductor composite on the substrate surface (discussions on the mechanism in Supplementary Information).

## Results

**Patterned deposition in situ**. The whole fabrication process is illustrated in Fig. 1b. Part A (the metallate) and Part B (the semiconductor nanoparticles) are chosen based on the material to be deposited. Thoroughly mixed solution of these two is drop-casted on the substrate surface. A home-built laser writing system is used to write the patterned material in-situ. While keeping the sample in the setup, the residue solvent is removed by a pipette, then the deposited material and substrate are cleaned with water. Examples from macroscopic scale down to microscopic scale of this LIMD method and their zoom-in details are shown in Fig. 1c (experimental details in Supplementary Information). On the left, we wrote the logo of The Chinese University of Hong Kong (CUHK) in platinum on a glass substrate. Making a similarly complicated structure in small scale using photolithography is difficult as either incorrect exposures or lift-off failure can ruin the fine details in the pattern. Since there are no development or lift-off steps, this complicated pattern came out straightaway.

Depositing patterns with two or more different materials, an important task in nanotechnology, is even more difficult for traditional fabrication methods, because multiple rounds of material depositions and photolithography with fine and accurate alignments are required. It is even harder to reach a good alignment among multiple rounds of material deposition within the whole area. However, it becomes a simple task with the LIMD method. To deposit different materials, above procedure just has to be repeated. In between two depositions, the reagent is changed without removing the sample from the setup. Thus, the alignment can be kept within the optical resolution of 400 nm. Here, as an example, in the centre of Fig. 1c, a Yin-Yang-fish symbol was written with gold and platinum. On the right of Fig. 1c, we drew a Chinese traditional ink painting with platinum and gold: the dark body parts of the panda were drawn in platinum, and the rest, including the contour of the panda and the bamboo, were drawn in gold. In both examples, with the LIMD method, these patterns were written with the least effort needed but high fidelity and submicron precision. Furthermore, by employing microfluidic chips with multiple channels in the future[13], depositions with different materials can be simplified further and automated.

**Universality of the LIMD method**. Owing to its universality in its mechanism, this method can be applied to a vast selection of materials and substrates. Based on the material to be deposited, the corresponding Part A component can be chosen. In Fig. 2, we show the collection of combinations we have tested. To be noted, this method works well with the commonly used noble metals, including gold, silver and platinum, as well as the widely used substrates, such as glass, quartz, sapphire and indium tin oxide

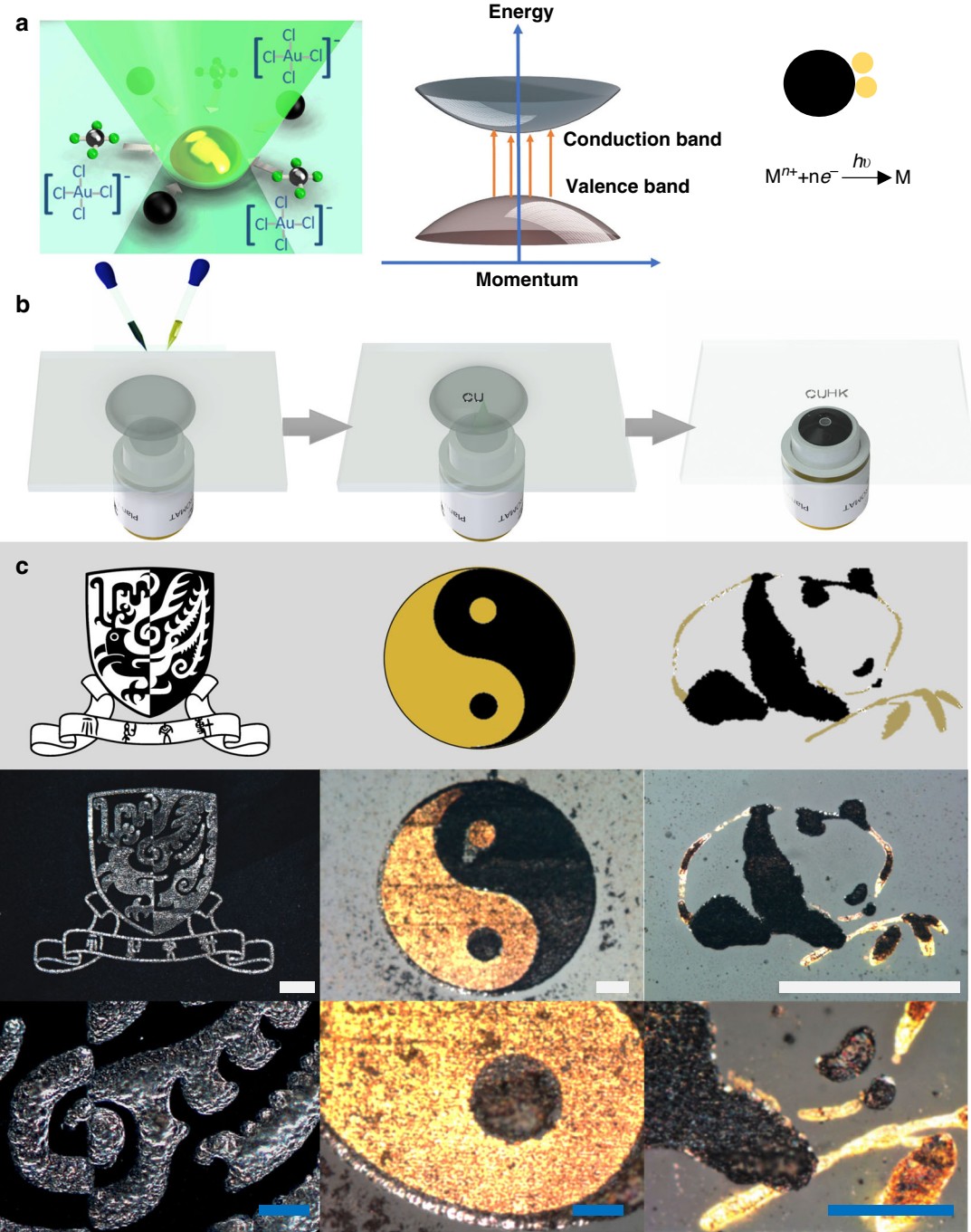

**Fig. 1 Laser-induced material deposition. a** schematic illustrations of the principle of the LIMD method. (left) Schematic illustration of the material deposition process, i.e., under the optical field, the semiconductor nanoparticles are trapped towards the focus and on their surface photo-induced reduction reaction converts the metal ion into metal deposition. (middle) Ideal band structure of semiconductor particles. A photon excites an electron from the valence band into the conduction band to become the free electron which triggers the photo-induced reduction reaction. (right) Schematic illustration of the photo-induced chemical reduction process. Free electrons excited by photons in the semiconductor particle (black sphere) reduce metal ions into metal particles (yellow spheres) on the surface of the semiconductor. **b** schematic illustrations of the experimental procedure. (left) Drop-casting the thoroughly mixed reagents on the substrate surface, with one pipette containing metallate and the other containing semiconductor nanoparticles. (middle) a 532 nm laser beam, focused by a microscope objective, creates deposition in the focal spot. (right) After the deposition and washing with water, the deposited pattern is left on the substrate. **c** three samples of depositions on glass slides done with the LIMD method. The designs are shown in the upper row. The images in the lower row are the detail zoom-ins of the middle row. (left) A logo of CUHK written in platinum. (centre) A yin-yang symbol written with platinum and gold. (right) A Chinese traditional ink painting written with platinum and gold. The white scale bars represent 50 μm, the blue ones represent 25 μm.

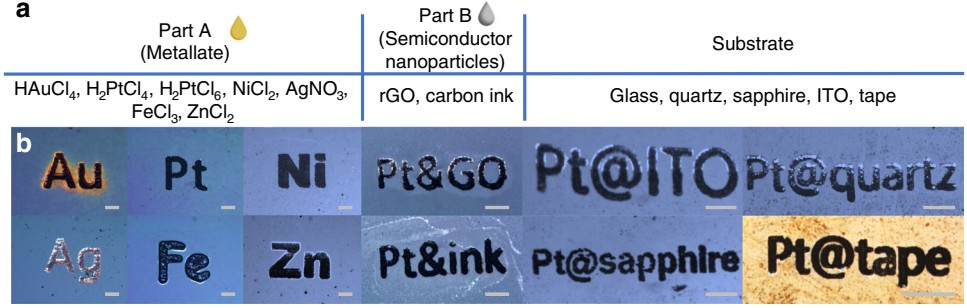

**Fig. 2 Material depositions with the LIMD method on various substrates. a** list of the ingredients and substrates used by the LIMD method in this work. **b** reflection type illumination microscopic images of different materials deposited on substrates. Except for the fourth one from the left in the upper row, which was done with reduced graphene oxide as Part B, the rest were done with carbon ink. Energy-dispersive X-ray (EDX) spectroscopy data, which show the element compositions in the layer are available in Supplementary Information. The grey scale bars represent 50 μm.

(ITO) (see Supplementary Information). It can also be used for depositing transition metals and substrates like flexible membranes and even tapes. This vast material possibility shows the potential of this LIMD method in general applications.

Part B in the recipe works as the reducing agent for the chemical reduction process. In general, all nanoparticles with a small band gap can work. Carbon based nanoparticles, for example, are a good choice. As the particles are required to mix with water based metallate solution, it is better to have them dispersed uniformly in water. Reduced graphene oxide fulfils these requirements and performs well. However, to our big surprise, the best performance was achieved using off-the-shelf carbon ink originally designed for fountain pens and Chinese ink brushes. After generations of optimisations by commercial companies, these ink particles not only are dispersed uniformly inside water, but have nearly identical hydrodynamic diameters down to 100 nm (see Supplementary Information). This feature gives the LIMD method even more advantages compared with traditional methods, such as cheap, easy to adapt and green to the environment.

**Quality characterisation of the deposited materials**. The quality of the deposition by the LIMD method can be evaluated from its physical structure and its physical performance. To estimate the spatial resolution reached by the LIMD method, both nano-dots and nano-wire are deposited by using a high numerical aperture (NA) oil immersion objective lens and fine tuned parameters. The scanning electron microscope (SEM) images of platinum dots with size around 0.5 μm, and that of a 1 μm wide iron oxide wire are shown in Fig. 3a, b, respectively. Also the cross-section of the iron oxide wire, cut with a focused ion beam machine, is shown in Fig. 3b. The deposition is uniform in size and solid inside. The surface roughness was ~30 nm, as measured by an atomic force microscope (AFM) (see Supplementary Information).

Noble metals are widely used as electrodes and wires in real applications owing to their excellent conductance. Thus, we performed electrical conductivity measurements (results shown in Fig. 3e), on a platinum wire deposited on a glass substrate (SEM image in Fig. 3c). After the LIMD deposition, the conductivity is ~10% of the bulk value of platinum. To examine the origin of this relatively low conductivity, the SEM image of its cross-section is shown in the insert in Fig. 3c. Unlike the cases in Fig. 3a, b, to reach the deposition in macroscopic scale in high writing speed here, low NA air objective lens and higher laser power are used. Under these conditions, the deposition consists of a stacking of pillars. This structure may come from the Turing instability[14]. The typical diameter of the pillar structure, which is around a few hundred nanometres, is mainly determined by the penetration depth of the light in the materials (see Supplementary

Information). Also, as this reaction–diffusion instability mechanism tends to deplete the resources in between reaction centres, the connections among pillar surfaces can be weak. This leads to the low conductivity. To solve this problem, the structure was sintered in air in the setup with the same laser[5]. The resulting conductivity rises to 18% of the bulk value (detail analysis in Supplementary Information).

With this LIMD method, not only conductors, but also insulators can be deposited. One approach is to use iron based metallate as Part A. Iron is so chemically active that even by using oxygen-reduced solution, the deposition ends up as iron oxide instead of pure iron. It turns out to be a good insulator as shown in the I–V curve measurement in Fig. 3f. The SEM image of the cross-section in Fig. 3d proves the structure is compact-packed. Therefore, this iron oxide deposition can be used as a practical insulator for general purpose.

Besides electrical conductance, we also calibrate their mechanical properties. By using nanoindentation method, the Young's Modulus of Pt deposition is determined to be 1.0 GPa. The number is similar to graphite which is the softer component in the deposition. On the other hand, the deposition is elastic and flexible due to this composite nature (detail analysis in Supplementary Information). This kind of high conductance and high flexibility composite material is ideal for making the flexible electronics.

The LIMD method can be used to deposit ferromagnetic materials as well. Take nickel as an example. We wrote four square shaped Ni layers with a permanent magnet polarising them during the deposition process, as shown in Fig. 3g. The magnetic profile then was measured with nitrogen vacancy (NV) centres inside nanodiamonds spread on the sample[15–18]. The splitting of electron spin resonance lines from these NV centres, shown in Fig. 3h, i, is due to the Zeeman effect of the residue magnetic field from the Ni layers. The field is ~30 Gauss inside the Ni square while close to none outside (details in Supplementary Information). This implies the ferromagnetic property in the deposited layer. Thus, this LIMD method enables us to in situ fabricate micromagnets and complicated magnetic structures.

**Applications by the LIMD method**. The performance of the deposited materials can be further demonstrated in three application devices. Flexible electronics and wearable technology are booming fields[19,20]. The commonly used flexible substrates usually have low melting points. Traditional deposition methods such as metal thermal evaporation may cause the melting of the sample surface. This LIMD method has a low working temperature (see Supplementary Information) and is water-solution based. It even works well with various thin tapes. Furthermore, unlike other metal contacts, this composite metal deposition has a

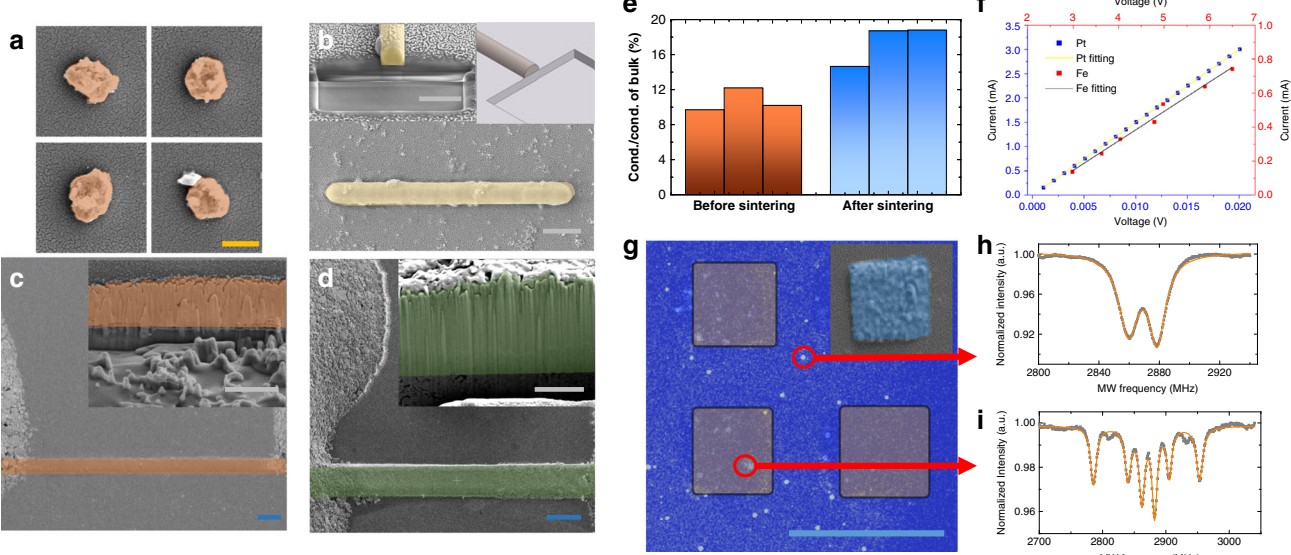

**Fig. 3 Physical properties of the deposited structures. a, b** SEM images show the spatial resolution reached by the LIMD method in depositing platinum dots **a** and iron oxide nano-wire **b** on glass slides. The schematic drawing of the deposition and the SEM image of cross-section view are shown as the inserts in **b**. **c, d** SEM images of the surface view and cross-section view (inserts) of platinum deposition **c** and iron oxide deposition **d** on glass slides. **e** The conductivity of platinum structures made before (orange) and after (blue) laser sintering. **f** The I–V curves measured for platinum (blue) and iron oxide (red) samples. **g** The fluorescence image from the confocal microscopy scan and the SEM image (insert) of nickel depositions on a glass slide. **h, i** the ESR spectra measured with NV centres in diamond in the middle point of the four nickel pads **h** and on top of one Nickel pad **i**. The grey scale bars represent 2 µm, the blue ones represent 50 µm, the yellow one represents 500 nm. Experimental details are in Supplementary Information.

conductance similar to metal while being flexible like a polymer. On the other hand, simplification in production and flexibility in design can be key factors for commercial developments. The direct laser writing nature of the LIMD method provides a simple and cheap way for production and customisation. As examples, we build two proof-of-principle devices: a resistive flex sensor and a resistive touch sensor. Resistive flex sensors are important parts for robotics[21]. The schematic drawing of this sensor is shown in Fig. 4a. A platinum line with length of 300 µm and width of 50 µm was written on a Kapton tape. The resistance of the wire shows a linear dependency with the curvature, as shown in Fig. 4d. The device reaches similar sensitivity compared to current sensors, but is two orders of magnitude smaller in size[21].

From this deformation-sensitive mechanism, resistive touch sensor devices can be further developed[22]. As shown in Fig. 4b, two parallel platinum squares were written on a Kapton tape. Depending on the location where pressure was applied on, the resistances of both lines show different trend of change, as shown in Fig. 4e. Based on this dependence, a measurement of both resistances can determine the location of the touching.

Besides applications in macroscopic scale, with this method also devices for microscopic applications can be fabricated with submicron accuracy. One of the major road blocks for nanotechnology is manufacturing high performance devices with high precision with respect to tiny or chosen samples[23]. The LIMD method provides a solution as the deposition and imaging are in situ. One key device in solid state based quantum information science is the microwave waveguide, which is used to enhance coupling of microwave radiation to solid state qubits[15,24]. This waveguide has to be in close vicinity to the tiny qubits and it should be able to transmit high power microwave signals. Shown in Fig. 4c, combining both confocal microscopy imaging and the LIMD system, we directly wrote a microwave waveguide near a nanodiamond particle. This waveguide performs as good as structures made with conventional method, as shown in the electron spin Rabi oscillation measurement (Fig. 4g).

With a cross-section of only $27 \times 2\ \mu m^2$, this structure works well for a large power range. Shown in Fig. 4g, is the linear dependency of the Rabi frequency to the square root of the applied microwave power up to 6.2 Watts. Furthermore, since the laser writing setup is compatible with other microscopy-based setups, it is possible to combine both the sample characterisation measurements and the LIMD method together. For example, in Supplementary Information, while depositing materials on a nanodiamond with the LIMD method, we simultaneously use the NV centres inside the nanodiamond as the thermometer to measure a possible heating effect. Methods like this will benefit the research fields where a big variation of the sample properties are unavoidable, such as two dimensional materials, nanoparticles and nanostructures[25–27].

With this LIMD method, not only 2D, but also 3D patterns can be made, owing to its mechanisms. By moving the focus in vertical direction, 3D material deposition can be done in the layer by layer manner. As an example, a 3D topographic map was written and is shown in Fig. 4i. Unlike other 3D laser writing methods with the requirement of high power ultrafast laser, a CW laser with ~100 mW is enough for the LIMD method. This makes this method one of the simplest methods for 3D laser writing, as well as a method which is practical in both 2D and 3D fabrication.

Reflow soldering is one of the most widely used bonding methods. We further demonstrated the reflow soldering working with the structures made by the LIMD method (see Supplementary Information). This paves the way for its application in modern integrated circuit industry.

One more special application of this LIMD method is repairing circuits. After the circuit has been fabricated, it becomes hard to repair a small crack or a broken pad with photolithography method. It is also difficult to reach high spatial resolution using the conductive epoxy. This LIMD method can provide a solution. We demonstrate this with an ITO structure commonly used for solar cell. A 20 µm crack was cut by a diamond cutter in the

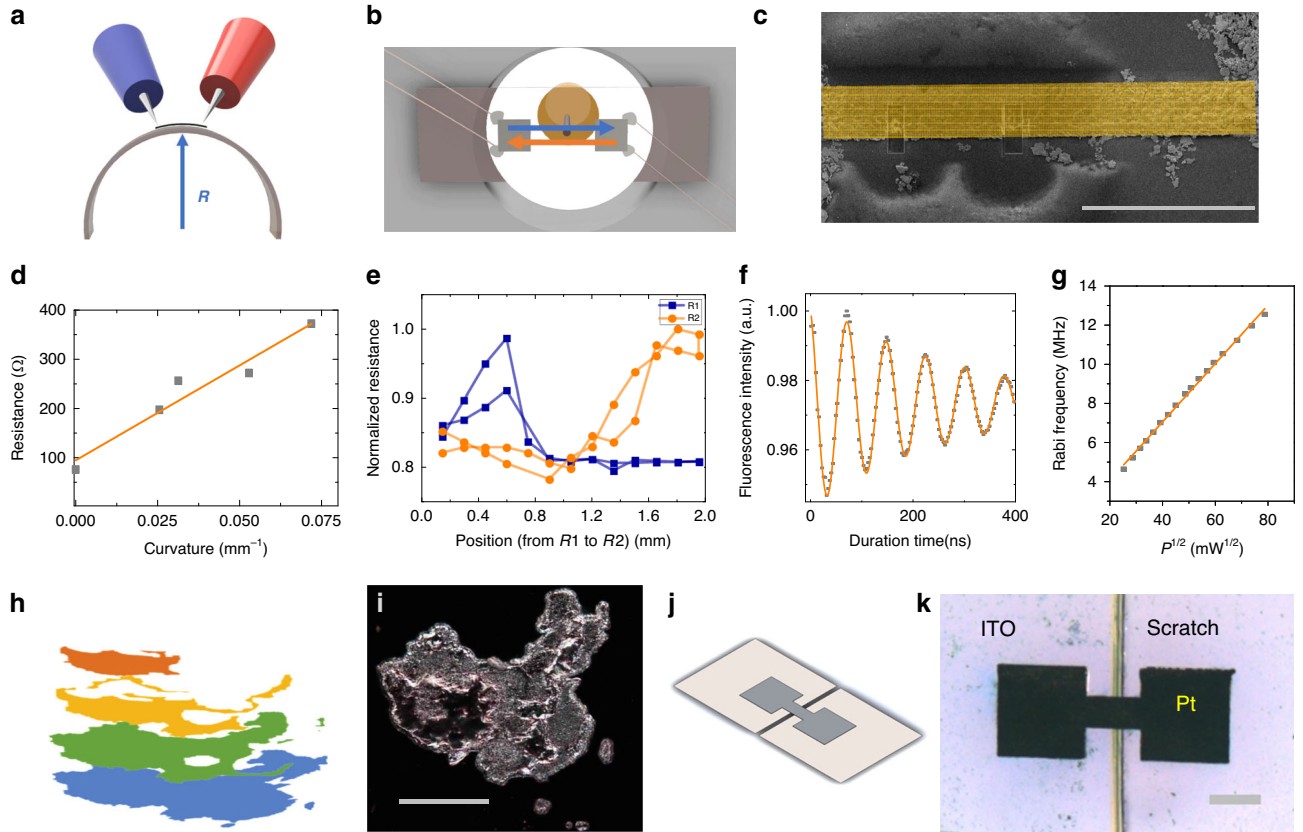

**Fig. 4 Devices made by the LIMD method for practical applications. a–b** schematic illustrations of two flexible devices: a resistive flex sensor **a** and a resistance-based touch sensor **b** (more details in Supplementary Information). **c** the SEM image of the microwave waveguide coupled with nanodiamond particles. **d** resistance dependence with curvature measured in the device in **a**. **e** resistance dependence of two platinum squares in the device **b** with position of touching. The resistance is normalised to the values without touching. **f** the microwave driven Rabi oscillation of electron spins in NV centres in diamond in the vicinity of the waveguide in **c**. **g** the microwave power dependence of the Rabi oscillation frequency. The microwave power was measured after the microwave power amplifier before entering the microwave waveguide. **h** design patterns for 3D laser writing. The structure was written in the layer by layer manner. The raw topography data were obtained from the Ministry of Natural Resources of China. **i** a 3D topographic map made by the iron oxide deposition. **j**, **k** the schematic design **j** and the microscopic image **k** of repairing the gap between ITO contacts with the LIMD method. The grey scale bars represent 100 µm.

middle of the ITO pad. As shown in Fig. 4j, k, a Pt bridge is made via the LIMD method to restore the connection (see Supplementary Information for details).

## Discussion
It is worth to mention that, compared with other conventional microstructure fabrication devices, this microscopy-based LIMD setup also reduces the requirements on equipment and training dramatically. An ink-jet laser printer for directly printing materials could even be made with this method. It would make the print-out of electrical circuits in the field possible, and reduce the fabrication costs in labour, material and equipment. After the deposition with the LIMD method, all the residue solution can be recycled to minimise material waste and pollution. Moreover, unlike photolithography method where the organic solvent often puts the sample surface in risk, there is no organic photoresist or solvent involved in this method. As current electrical fabrications produces enormous amounts of waste, including organic solvents, chemical etching wastes and left-over materials, which endanger the whole environment, this nearly waste free method will play a big role in reforming the current industry towards a greener one.

In summary, we introduce a patterned material deposition method based on the combination of photon-induced chemical reduction and laser-induced optical trapping with semiconductor nanoparticles as the agents. In this first demonstration, we already showed the successful applications with a vast selection of materials on a large selection of substrates. Much more recipes can be designed based on this general mechanism. For example, the nanoparticles can be inorganic semiconductor or perovskite structures, or other organic materials. With more experts from different disciplines joining, the LIMD method will be developed into a powerful tool in the modern material research.

## Methods
**Chemical and materials**. Gold(III) chloride hydrochloride (HAuCl$_4$ 99.995%), Nickel chloride (NiCl$_2$, ≥98.0%) and Silver nitrate (AgNO$_3$, 99.9999%) were purchased from Sigma-Aldrich. Chloroplatinic acid hydrate (H$_2$PtCl$_6$ · xH$_2$O, ≥99.9%) was purchased from Jiangsu Hanggui Catalyst Co., Ltd. Zinc chloride (ZnCl$_2$, ≥98.0%) was purchased from Tianjin Yongda Chemical Co., Ltd. Iron(III) chloride hexahydrate (FeCl$_3$ · 6H$_2$O, ≥99%) was purchased from Tianjin Guangfu Chemical Co., Ltd. Reduced graphene oxide powder (purity >99%) was purchased from Hangzhou Hangdan Optoelectronics Technology Co., Ltd. Carbon inks were purchased from suppliers Shanghai Hero Pen Company (No. 234 black ink) and Sailor Pen (Nano Kiwa-guro Ink, ultra-black). Glass slides 25.0 × 75.0 mm 1.0–1.2 mm thickness was purchased from Lab'IN Co. and 22 × 22 mm 0.13–0.16 mm thickness was purchased from Paul Marienfield GmbH & Co. KG. Quartz plate (20 × 20 × 1 mm) was purchased from Lianyungang Weida quartz Co., Ltd. Sapphire plate (D8 × 1 mm) was purchased from Shengyakang Optics Co., Ltd. ITO plate (380 nm/1.1 mm thickness of ITO/glass) was purchased from South China Science & Technology Co., Ltd. Kapton tape (50 µm thickness) Mileqi Adhesive Co., Ltd.

**Preparation of metallate solution and semiconductor nanoparticle suspension**. Gold(III) chloride hydrochloride ($HAuCl_4$), zinc chloride ($ZnCl_2$), nickel chloride ($NiCl_2$), chloroplatinic acid ($H_2PtCl_6$), silver nitrate ($AgNO_3$) and iron(III) chloride ($FeCl_3$) were dissolved in Milli-Q water separately, and the solutions were sonicated 15 mins to obtain different stock metal solutions. Their stock concentrations are the following: $AgNO_3$ 30 mmol/L, $NiCl_2$ 50 mmol/L, $FeCl_3$ 50 mmol/L, $ZnCl_2$ 200 mmol/L, $H_2PtCl_6$ 30 mmol/L, $HAuCl_4$ 2.5 mmol/L. Reduced graphene oxide and carbon ink were dissolved in Milli-Q water separately, and the solutions were sonicated 20 mins to obtain different stock of reagents. Their stock concentration is the following: 1. Carbon ink (from Hero Pen Company) 1:100. 2. Reduced graphene oxide 1 mg/mL. Dynamic light scattering measurements of the reducing agents suggest that the average hydrodynamic diameters of graphene oxide and two types of carbon inks solution are 710 nm, 90 nm and 160 nm, respectively (results please refer to Supplementary Information Fig. S4). The mixture of different combination of metallate solutions, semiconductor nanoparticles and ethanol (with volume ratio 1:1:2) were sonicated for 15 s before laser irradiation. The ethanol is meant to lower the surface tension and disperse the bubble produced during the LIMD process. The concentration of metallate solutions and semiconductor nanoparticles for material deposition on different substrates varies depending on the application. The concentration of carbon ink in each solution is 1/6000. The final working concentrations of the metal ions are: $AgNO_3$ 7.5 mmol/L, $NiCl_2$ 12.5 mmol/L, $FeCl_3$ 12.5 mmol/L, $ZnCl_2$ 50 mmol/L, $H_2PtCl_6$ 7.5 mmol/L, $HAuCl_4$ 0.625 mmol/L. The concentration of reduced graphene oxide in each solution is 0.25 mg/mL.

**Laser writing system**. The experimental setup is a home-built direct laser writing system. The system contains a CW laser wavelength at 532 nm (DPSS laser from Laser quantum), an acousto-optic modulator (AOM) (Gooch & Housego 3350-199), two axis Galvo scanner (Sino Galvo JS2205), a $20 \times 0.75$ NA air objective lens (Nikon) or a $100 \times 1.3$ NA oil objective lens (Zeiss), sample holder (three axis manual stage from Thorlabs) and a CCD camera (IDS Imaging Development Systems GmbH).

**Laser-induced material deposition**. In the following, the deposition process of platinum on glass is described. A glass substrate covered with an aqueous solution of the mixture of $H_2PtCl_6$ and carbon ink was subjected to the laser writing system. A CW laser beam was focused at the substrate/liquid interface to introduce the reduction reaction between metallate and semiconductor nanoparticles as well as induce optical trapping. The laser on/off status was controlled by an AOM, while the transmission, and luminescence/scattering at the laser focus were imaged by the same microscope. The LIMD parameters for large structures were the following: the exposure laser power was 60.4 mW measured in front of the objective lens. The exposure time per pixel was 30 ms, and the pixel size was 1.17 μm. For microstructure deposition (Fig. 3a, b), the exposure parameter was 0.485 mW power and 1s exposure time. After the exposure, the residue solution was taken out by a pipette. Further cleaning was performed by adding and taking out pure water solution two–three times. All the procedures had been done without taking out the sample from the LIMD system. When deposition of a second type of material is needed, the procedure, described above, can be repeated using a different solution. The more detail of exposure parameters please refer to Supplementary Information Note 3.

**Image acquisition**. The optical microscope images are acquired by an Olympus BX60 Micrscope with 5×, 10×, 20×, 50× and 100× UMPlanF objectives and a ZEISS AxioCam MRc5 microscope camera, controlled by AxioVision 4.0 software, for image acquisition. The first column (from left to right) of Figs. 1c and 4i are acquired by dark field mode and the second and third column of Fig. 1c, all in Figs. 2b and 4k are acquired by bright field mode.

The SEM images are acquired by FEI (ThermoFisher Scientific) Scios2 Dual Beam using 5.00 kV acceleration voltage and 5.0 spot size. The milling of deposition for cross-section exposure is performed by the same device using 30 kV acceleration voltage and Si as the ion beam source. In all, 7 nA milling current is first applied to expose the cross-section and 0.3 nA is applied to clean the milling residue on the cross-section afterwards. All samples were coated a thin layer of gold (POLARON SC502 Sputter Coater) before entering SEM chamber.

**Deposition characterisation**. The electronic property of deposition is measured by Keithley 2400 with probe station. Silver glue (EPO-TEK H20E) is used for electrode as contact between deposition and probes. The elemental constitutent of deposition is measured by energy-dispersive X-ray spectroscopy (EDAX. Inc.). The nanoindentation was performed under Veeco diDimension icon AFM. The AFM probe is OMCL-AC240TS-R3 with 2 N/m spring constant from Olympus. Measurement involving NV centres (remanent field of nickel pads Fig. 3g–i, microwave waveguide Fig. 4f–g) were performed by a home-made confocal microscope setup that contains a 520 nm laser (PL520, Thorlabs), a $100 \times 0.95$ NA air objective lens (Olympus), a 3D piezo nanopositioner (Physik Instrumente GmbH), a 650 nm longpass filter (DMLP650) and an avalanche photodiode (Excelitas Technologies Inc.).

## Data availability
The images and data for plots in the main text are available at Zenodo with the identifier https://doi.org/10.5281/zenodo.4032439. Additional data are provided in Supplementary Information and are available from the corresponding author upon reasonable request.

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

## Acknowledgements
We thank J. Hofkens, M. Totzeck, J. Wrachtrup and W. Yao for the fruitful discussions. We thank S.K. Li, C.H. Lai and M.H. Yeung for the technical helps. S.Y. acknowledges the financial support from CUHK start-up grant and Hong Kong RGC (GRF/ECS/24304617, GRF/14304618 and GRF/14304419).

## Author contributions

K.X. and S.Y. conceived the original idea, designed the experiment and wrote the paper, Yf.C., S.F.H., W.K.L., Y.C., Y.S., K.X., S.Y. performed the experiment and analysed data, Yf.C., S.F.H., W.K.L., Y.C., Y.S., K.K., S.J., K.X., S.Y. commented on the manuscript.

## Competing interests

Yf.C., S.F.H., K.X., S.Y. and The Chinese University of Hong Kong have filed a patent application related to the photon-induced material deposition (European Patent Application no. 19218640.1).
