## [Peer Review File · Nature Communications]

Reviewer #1 (Remarks to the Author):

Yang and co-workers proposed a method for patterning diverse materials on diverse substrates using laser-induced chemical reduction and trapping. They demonstrated this fabrication technique for metals, insulators and magnetic materials. They also presented applications of this fabrication technique by fabricating structures in 3D along with *in situ* characterization owing to the benefits of laser-induced process. However, enough characterization is not performed to determine the printing quality with respect to the existing techniques. Also, several laser-based fabrication techniques have been proposed, both using cw lasers and fs lasers, which have demonstrated the versatility of materials used for printing and different applications. Although the demonstrated mechanism is new, it does not suffice the novelty standards that are usually set for Nature Communications. For this reason, I recommend rejecting this article. Please find my concerns attached.

Major reviews:

1. One of my major concerns is the printing quality. If the metal particles are attaching to the semiconductor particles which act as bonding agents, it means that the patterning involves both semiconductor and metallic particles. The authors must explain more on how they could achieve patterning a single material uniformly.
2. Also, a claim is made that iron oxide deposition can be used as a practical insulator, the semiconductor particles also affect the insulation.
3. A comparison must be made between the existing patterning mechanisms and the current technique.

Minor reviews:

1. How is the reagent cleaned/changed from the sample without removing the sample from the setup, without altering the alignment of the sample?
2. Order of 1c looks like an example for microchannel fabrication which is misleading. The phrasing must change, or it must be before the microchannel sentence.
3. Fig 3b is slightly difficult to understand. Also, could you explain what difference between figs. 3b is and 3d.
4. Figure 3e label is wrong, it must be Cond/Cond of bulk

Reviewer #2 (Remarks to the Author):

In the report, the authors demonstrated micro-deposition of materials by spatially selected photoinduced reduction combined with laser tweezer and the fabrication technique is named light induced material deposition (LIMD) method by the authors. The point of the method is that nanoparticles with small band-gaps are used as so-called photo-sensitizers for reduction of metallic ions in aqueous solution and laser tweezer can trap the nanoparticles and increase their concentration for improving the chemical reactivity of the photoreduction in the focal point of the laser. The method proposed in the report looks versatile; the authors demonstrated micro-deposition of noble metals, oxide, and ferromagnetic metal. In addition, they also fabricated by using LIMD method flexible devices such as flex and touch sensors.

I think the results shown in the paper is interesting and topical then it can potentially be publishable in Nature Communications after appropriate revision. My individual concerns are shown below.

(1) Although important and pioneering works are cited in the introductory part, two important reports are missing in the context of the combination of photoinduced chemical reaction and optical trapping: (i) Tamitake Itoh, et al., "Optical force enhanced by plasmon resonance allowing position-sensitive synthesis and immobilization of single Ag nanoparticles on glass surfaces", *Applied Physics Letters*, Vol. 94, 144105 (2009) [<https://doi.org/10.1063/1.3106617>] and (ii) Syoji Ito, et al., "Confinement of Photopolymerization and Solidification with Radiation Pressure", *Journal of the American Chemical Society*, Vol. 133, 14472–14475 (2011) [<https://doi.org/10.1021/ja200737j>]. These two reports positively and successfully utilized optical force for micro-deposition/micro-fabrication by photoinduced chemical reactions.

(2) In the introductory part, the authors claim that the complex process of lithography prevents the wide spread of quantum beam lithographic techniques. However, the lithographic nano/micro-fabrication techniques are now widely used in a lot of fields ranging from a laboratory level to mass production in factories. Difficulties have been resolved by the great effort of scientists and engineers although the process is actually complex. I suggest to the authors that they rephrase the point in a respectful form.

(3) The description of Fig. 1a and 2b are missing and the illustration of Fig 1a left is misleading. The green spiral line in Fig. 1a shows circularly polarized beam? In the presence of light field, the "sensitizer nanoparticle" reduces metallic ions. But in Fig. 1a left, some of nanoparticles are already covered with reduced metal despite lack (or very small number) of photons outside the focus. The illustration in the middle of Fig. 1a gives little information (description) of the deposition mechanism. Give Fig. 1b words describing important stuffs, e.g., the contents of the two pipettes, green circular cone (this is maybe laser beam from the bottom), etc.

(4) The 1st line of RESULTS, what are Part A and Part B?

(5) Discussion of the mechanism of the deposition is missing. In a related point, the stiffness of the deposition is also very important in the application of the method. The evaluation of the adsorption stiffness and mechanical strength improve the quality of the paper.

(6) The photoexcitation by the laser produces an exciton in the nanoparticle and the excited electron can reduce a metal ion near the particle. At the same time, the photoexcited electron relaxes through a non-radiative pathway to generate heat. Especially, when carbon nanoparticles are used, I think the photothermal effect is not so small. The discussion on the photothermal effect is useful for readers who will use the method for heat-sensitive substrates/films of organic compounds.

(7) From the view point of "nano" fabrication, spatial accuracy of the material deposition method is crucial. It should be evaluated and discussed.

Dear Editor and Reviewers,

We thank you for your efforts and your time. Your comments are extremely important and helpful for us. Please find below your queries in green font and our response in black font.

Yours truly,

The authors

Reviewer #1 (Remarks to the Author):

Yang and co-workers proposed a method for patterning diverse materials on diverse substrates using laser-induced chemical reduction and trapping. They demonstrated this fabrication technique for metals, insulators and magnetic materials. They also presented applications of this fabrication technique by fabricating structures in 3D along with *in situ* characterization owing to the benefits of laser-induced process. However, enough characterization is not performed to determine the printing quality with respect to the existing techniques. Also, several laser-based fabrication techniques have been proposed, both using cw lasers and fs lasers, which have demonstrated the versatility of materials used for printing and different applications. Although the demonstrated mechanism is new, it does not suffice the novelty standards that are usually set for Nature Communications. For this reason, I recommend rejecting this article. Please find my concerns attached.

Response:

We thank the Reviewer for sharing his/her concerns with us. These suggestions help us polishing this work. Following these suggestions, we have systematically measured the electrical and mechanical properties on the printed layers, and compared with the existing techniques (details in below). We would like also thank the Reviewer to point out that the deposition should be called as material composite. We made the modifications throughout the work.

Here we would like to address the novelty problem raised by the Reviewer. There are two directions of novelty: the mechanism and the applications this method leads to.

1. The method and mechanism.

One of the uniqueness, as the Reviewer kindly pointed out, is the mechanism of this LIMD method: by using semiconductor nanoparticles as the photon-induced reduction agent and objects for optical trapping, we can combine the optical trapping and the light induced reduction together to make a universal deposition of solid material composite. In some sense, it is similar to the invention of the concrete by adding pebbles in cements. Cement is weak and was used only to hold bricks together. But by introducing stones and later metal frames, this composite material revolutionized the civil engineering.

Traditional methods, such as photolithography, 3D printing and some of direct laser writing methods, are good at shaping and depositing polymer based materials with good quality and in a large range of scales. However, for metals or other inorganic materials, not many universal methods exist in practice. Therefore, up to now, most fabrications are still based on using patterned polymer as masks. The whole process requires

expensive equipment, high quality facility, experienced users, requirements for substrates and tedious steps.

There are other approaches. For example, metal composites can be made by mixing metal particles with polymer, such as, silver paste and silver epoxy. The conductivity is several orders of magnitude lower than the bulk materials. High power pulsed laser can either melt or vaporize metals, but this has many limitations like dangerous temperature for substrates, poor resolution and the bulky/expensive/dangerous equipment. There are chemical based methods or using heating to create metal nanoparticles. By combining with methods like the optical trapping or heat annealing, people create micron size structures. But without good binding methods, it is hard to create macroscopic metal structure that can compete with the traditional lithography in properties. The state of arts in this field were demonstrated recently in two works: direct optical deposit inorganic nanomaterials via photo-chemistry reaction and then anneal the finished structure (Science, 357, 385-388, 2017). However, the thickness of layers can only be as thick as 100 nm due to the limited penetration depth. In the other work, scaffold of hydrogel were made, functional materials were attached, then the frame was dehydrated, and what left-out was the deposition (Science, 362, 1281-1285 2018). In both methods, complex chemistry treatment/synthesizing and multi-step preparation and development are needed. It is still far from being single step fabrication.

(We review the existing laser based fabrication techniques in details and highlight the uniqueness of this LIMD method in the section below for answering the Reviewer's question.)

Just like adding pebbles or metal frames to cements, by adding semiconductor nanoparticles, we create a single-step method not only powerful for applications as explained below, but also bringing a new direction of research. Because this working mechanism is universal, researchers from different areas, like chemistry, physics and material science, can design new ingredients by changing the recipe. For example, the semiconductor particles can be selected from inorganic materials to organic based solar cell materials; the materials to be deposited and the metallates also have a huge range of selection. Furthermore, this method may not just limit to material deposition. It can also be used for solving other tricky problems like the material etching.

To illustrate the uniqueness and shine light for the wide adaption, we have also added a new chapter (Chapter 5, Page 28-38) in the SI on the mechanism with systematical analysis and experimental data.

2. The applications.

Another importance of this work is on the usage of this method. So many methods have been developed, however, there are limited methods that challenges the dominance of the photolithography. By performing demonstration, characterization, and building devices, we would like to convince the audience of the potential of this LIMD method.

Just for the first try, we have demonstrated the deposition of various materials and substrates, already one of the most versatile methods; the spatial scales in both the size and thickness can be from sub microns to macroscopic; the multiple material deposition becomes accurate and simple.

As correctly pointed out by the Reviewer, the finished material is a composite of metal and nanoparticles. The electrical and mechanical properties are still comparable to the bulk values and in par with other laser writing methods (we added detail study in the revised SI, Chapter 4.7-4.8, Page 18-28). Since this LIMD method greatly reduces the working temperature and simplifies the fabrication steps and requirements, this is a valuable trade-off. Furthermore, the deposited materials have high flexibility due to their composite nature. Thus, it is a unique material with conductance similar to bulk metals with flexibility like Nylon. It is an ideal material for flexible devices.

Besides making fancy pictures, we would like to seriously demonstrate that this method is ready for the practice. (Based on our limited knowledge, among all papers we read, this work is one of the rare ones that demonstrate the practical applications.) We worked out several practical devices. There are several highlights: 1. The method can reach sub-micron resolution while the whole structure can be in big scales (We have fabricated structures several centimeter in sizes), it can deposit easily several micrometer thick layers, which can be tricky for other methods; 2. This method can be used together with other optical based measurements, so it is ideal for precision fabrications for quantum devices in the quantum information science, but not limited to it; 3. It has low working temperature, good flexibility and is well suited for flexible devices; 4. It is compatible with industry techniques like the wire bonding.

Moreover, in this revision, we added a new application: we can use this LIMD method to repair circuit boards. This was kindly pointed out by Prof. Dr. M. Totzeck from Zeiss. It is currently a big problem in industry that once there is a small crack or a broken pad in the circuit board, it is nearly impossible to repair via photolithography. Using the silver epoxy is hard for tiny cracks and the conductance is low. The LIMD method provides the best solution in this situation so far. In this revised manuscript, we have added examples of using the LIMD method to repair electronic circuits with down to micron-sized gaps. As far as we know, this is the solution no other method can deliver.

Thus, the method we present in this work has great potentials in the development of the modern technology.

This LIMD method also has another bonus. This single-step method is much easier to use and to adapt. Compared with other methods, this one can be nearly expert-free. A high school student can build one at home. The performance can be as good as from a professional fabrication lab. It is nearly waste-free, i.e., no metal waste, no organic waste, and good for the environment. If the ink-jet style material printer can be made eventually, it can make a boost worldwide like 3D printers did.

Overall, this LIMD method is important for future applications and the novel mechanism will bring a new field of designing fabrication techniques.

Below, we also addressed both major and minor concerns brought by the Reviewer. We hope these explanations and experimental results can answer all the concerns. Also, in this revised SI, we double the content with detailed study of the quality as well as the mechanism of the deposition. This revised manuscript not only shows a novel method, but also gives a systematic and rich study of it. We hope that the Reviewer can be convinced on the novelty and the importance this work demonstrated.

Major reviews:

1. One of my major concerns is the printing quality. If the metal particles are attaching to the semiconductor particles which act as bonding agents, it means that the patterning involves both semiconductor and metallic particles. The authors must explain more on how they could achieve patterning a single material uniformly.
2. Also, a claim is made that iron oxide deposition can be used as a practical insulator, the semiconductor particles also affect the insulation.

Response:

We would like to thank the Reviewer for pointing this important point. The finished deposition is a composite made of semiconductor nanoparticle coated with metal/insulator, not a pure metal/insulator. We modify the manuscript to clarify this.

The uniformity is achieved due to the working principle: each semiconductor nanoparticle works as the agent for photo-induced reduction. The material to be deposited accumulates on the nanoparticle surface until the optical penetration depth is reached. This deposition also can bond other particles to form thicker layers. We add a new section in SI (Chapter 5, Page 28-38) to illustrate this process in detail with the supporting data from experiments.

As the Reviewer pointed out, compared with pure materials, this composite material has some compromises in its material properties. For example, the conductivity of metal is less than that of the pure bulk material. As shown in the paper, the conductivity of Pt composite is around 18% of pure Pt. To be noted, the state-of-art before this work is, a silver deposition with a conductance of 13% of bulk silver (Science, 362, 1281-1285 (2018)).

Actually, as pointed out by the Reviewer, a big part of the reduction in conductance compared with bulk is due to the carbon particle inside the composite. As estimated in SI (Chapter 4.7 Page 18-23), with this carbon concentration, the ideal conductance of this composite is 42.5% of pure Pt. Thus, we reached nearly 44% of the ideal value. Similar situation for the iron oxide. And these values have rooms for improvement by optimizing the recipe.

Nevertheless, we think this is still a valuable trade-off for the features we illustrate in the previous answer to the question.

Furthermore, this composition nature brings advantages as the materials enjoy the features from both components. As suggested by both Reviewers, we characterized the mechanical properties of the deposition (SI, Chapter 4.8, Page 23-26). It has similar Young's Modulus as graphite, because the component of carbon ink is the softer part of the deposition. Thus, the material shows great elasticity and flexibility, in par with materials like nylon. This makes the composite unique as it provides both high conductivity like noble metals and a flexibility like nylon. This is a promising material for making flexible devices.

3. A comparison must be made between the existing patterning mechanisms and the current technique.

Response:

We thank the Reviewer for the suggestion. We enlarge the content of comparison in the introduction part of the manuscript. Here, we would like to explain more in details.

Currently there are many patterning techniques. They are mainly summarized to six types. Fused deposition modeling (FDM), polyJet printing, direct ink writing (DIW), stereolithography (SLA)/ digital light processing (DLP), selective laser sintering (SLS), and direct laser writing (DLW).

FDM, the common method in 3D printing, is based on the melting of polymer and is simple to use. However, it suffers from low resolution, large roughness, and lack of complexity. The PolyJet is based on material jetting, where liquid photopolymers are dropped and cured with UV light. It is able to do multi-material deposition with selected materials and good spatial resolution. However, the quality of metal deposition is far away from the bulk. Also the cost of the machine is high. DIW creates depositions by taking advantage of the high viscosity of the ink. But the deposited materials are mainly limited to polymer based materials, that have low resolution and are relatively fragile. SLA/DLP cures the polymer with light to form the structures. The method requires resin, resulting in the limitation of the material that can be applied. SLS uses high power laser to sinter material particles to form structures. However, it has issues like large roughness, limited materials to be used and the high costs of the device which requires high power lasers. Furthermore, most of these methods can only go down to sub-mm resolutions and the machines are expensive and specialized.

Most commonly used method is the DLW. It is the patterning of photoresist or photocurable polymer. It has high resolution and no need for additional support. However, the cost of the device, photoresist etc., is high, the material is limited to only polymer basis. If different materials are needed, material evaporation/sputtering, plating/lift-off are required. There are many new methods in the developing stages. As mentioned in the previous section, there are chemical based methods or using heating to create metal nanoparticles. By combining them with methods like optical trapping or heat annealing, people create micron size structures. But without good binding methods, it is hard to create macroscopic metal structures that can compete with the traditional lithography in properties. The state of arts in these fields were demonstrated recently in two works: direct optical deposit inorganic nanomaterials via photo-chemistry reaction and then anneal the finished structure (Science, 357, 385-388, 2017). However, the thickness of layers can only be as thick as 100 nm due to the limited penetration depth. In the other work, scaffold of hydrogel were made, functional materials were attached, then the frame was dehydrated, and what left-out was the deposition (Science, 362, 1281-1285 2018). In both methods, complex chemistry treatment/synthesizing and multi-step preparation and development are needed. It is still far from being single step fabrication. Also, there are few practical applications demonstrated with these methods.

Our LIMD method is a huge step forward from DLW. Especially, since it covers the deposition of inorganic materials which are the weak area of the traditional DLW methods. From the mechanism point of view, the nanoparticle trapped by optical force

works as the anchor for starting the deposition, and it also works as the agent for triggering the photon induced reduction reaction to glue particles and grow them into rigid structure. Thus, it solves problems existing in other methods. The deposition has electronic qualities similar to bulk metals and is elastic and flexible like nylon. It is able to deposit multi-material with precise spatial accuracy. It can have a spatial resolution around 500 nm and the whole structure size can be beyond centimeters (no limitation in the maximum size in principle); it can deposit easily several microns thick layers, which is tricky for some other methods. It is a method with low working temperature and can be organic free. Thus, it can be applied to organic substrates, and is well suited for flexible devices. Also, without the need of further development with organic solvents, this method is truly a single-step process.

The whole setup is compatible with optical measurements, so it can be used in *in-situ* deposition during sample characterization. Since in quantum technologies, it is crucial to construct quantum devices and quantum chips in selected sample areas with special physical requirements, this LIMD method provides the unique tool to address this challenge. Furthermore, compared to all the methods, the LIMD is a noncontact method. It has the possibility to perform under complex environments such as the repair of microelectronics. In this revised manuscript, we have added the example of using the LIMD method to repair micron sized gap in an electronic circuit (Fig. 4 j, k, and page 12, paragraph 2 in the manuscript; Chapter 8, Page 49-50 in the SI). As far as we know, this is a solution no other method can deliver.

The cost of the device as well as the cost of the reagent are low. On top of it, with microfluid channels, it is possible to recycle the raw material, which further reduces the cost.

Minor reviews:

1. How is the reagent cleaned/changed from the sample without removing the sample from the setup, without altering the alignment of the sample?

Response:

We thank the Reviewer for pointing out this insufficient explanation problem. Cleaning/changing solution without removing the sample is one of the uniqueness of the LIMD technique. Compared to the conventional photoresist-based lithograph technique, the LIMD does not require special treatment of the reagent, such as spin coating/soft baking/post baking/developing. After the material deposition, the residual reagent is simply taken out by the pipette. The sample is washed several times with DI water by the procedure of drop cast and taken out with the pipette. The whole procedure has been done without taking out the sample of the laser writing system. When a second material deposition is required, we simply drop cast a new type of the reagent on the sample and continue the material deposition. It indicates that a pattern with a different material does not require additional alignment.

We added this explanation into the SI as a new section (Chapter 3.4), Page 9-10.

Figure 1 Schematic drawing of the steps of changing solutions.

Also, as discussed in the next comment, in principle, microfluid technique can be combined with this LIMD technique. This can make the cleaning and reagent exchange automatically. Meanwhile, all the residual solution in the LIMD can be recycled, indicating nearly no waste of solution and pollution.

2. Order of 1c looks like an example for microchannel fabrication which is misleading. The phrasing must change, or it must be before the microchannel sentence.

Response:

We thank the Reviewer for pointing out this confusion. We exchanged the order of the description in the manuscript as the Reviewer suggested.

3. Fig 3b is slightly difficult to understand. Also, could you explain what difference between figs. 3b is and 3d.

Response:

We thank the Reviewer for pointing out this problem. We redesign the Figure 3 for better clarity as shown below. We add a group of nanodots made by the LIMD method as the new Fig. 3 (a) and combine old Fig. 3 (a,b) as the new Fig.3 (b). Together, these two new figures demonstrate the spatial resolution this method reaches.

The difference between Fig. 3 (a,b) and (c,d) is: to reach the best resolution, a high NA oil objective lens and a slow writing speed were used for making structures shown in (a,b). The cross-section image in the insert in (b) shows the interior structure is nearly perfect. On the other hand, to make practical structures in macroscopic scales, a low NA air objective lens, higher laser powers and fast writing speeds were used for making structures shown in (c,d) as well as most other devices in this paper. In this situation,

there are pillar like structures inside the deposition. As shown in the paper, sintering helps to improve the quality after the deposition in this scenario.

We also add a schematic drawing as the new insert to show the relationship between the cross-section SEM image (the inserts in Fig.3 (b,c,d)) and the SEM of the depositions (Fig. 3 (b,c,d)).

We revise the descriptions in the main text and figure caption to include these information and clarify the difference.

Figure 2 (Revised Figure 3 in the main text.) **Physical properties of the deposited structures.** a, b, SEM images show the spatial resolution reached by the LIMD method in depositing platinum dots (a) and iron oxide nano-wire (b) on glass slides. The schematic drawing of the deposition and the SEM image of cross-section view are shown as the inserts in (b). c, d, SEM images of the surface view and cross-section view (inserts) of platinum deposition (c) and iron oxide deposition (d) on glass slides. e, the conductivity of platinum structures made before (orange) and after (blue) laser sintering. f, the I-V curves measured for platinum (blue) and iron oxide (red) samples. g, the fluorescence image from the confocal microscopy scan and the SEM image (insert) of nickel depositions on a glass slide. h, i, the ESR spectra measured with nitrogen vacancy (NV) centres in diamond in the middle point of the four nickel pads (h) and on top of one nickel pad (i). The grey scale bars represent $2\ \mu\text{m}$, the blue ones represent $50\ \mu\text{m}$, the yellow one represents $500\ \text{nm}$. Experimental details are in Supplementary Information.

4. Figure 3e label is wrong, it must be Cond/Cond of bulk.

Response:

We thank the referee to point it out. We modified it in the revised manuscript.

Reviewer #2 (Remarks to the Author):

In the report, the authors demonstrated micro-deposition of materials by spatially selected photoinduced reduction combined with laser tweezer and the fabrication technique is named light induced material deposition (LIMD) method by the authors. The point of the method is that nanoparticles with small band-gaps are used as so-called photo-sensitizers for reduction of metallic ions in aqueous solution and laser tweezer can trap the nanoparticles and increase their concentration for improving the chemical reactivity of the photoreduction in the focal point of the laser. The method proposed in the report looks versatile; the authors demonstrated micro-deposition of noble metals, oxide, and ferromagnetic metal. In addition, they also fabricated by using LIMD method flexible devices such as flex and touch sensors. I think the results shown in the paper is interesting and topical then it can potentially be publishable in Nature Communications after appropriate revision. My individual concerns are shown below.

Response:

We thank the Reviewer for the high evaluation of our work. We are grateful for his/her expert opinions and kind guidance in this review. These suggestions guide us in the history of the field, better understanding of the detail process and better way to present the work.

(1) Although important and pioneering works are cited in the introductory part, two important reports are missing in the context of the combination of photoinduced chemical reaction and optical trapping: (i) Tamitake Itoh, et al., "Optical force enhanced by plasmon resonance allowing position-sensitive synthesis and immobilization of single Ag nanoparticles on glass surfaces", Applied Physics Letters, Vol. 94, 144105 (2009) [<https://doi.org/10.1063/1.3106617>] and (ii) Syoji Ito, et al., "Confinement of Photopolymerization and Solidification with Radiation Pressure", Journal of the American Chemical Society, Vol. 133, 14472–14475 (2011) [<https://doi.org/10.1021/ja200737j>]. These two reports positively and successfully utilized optical force for micro-deposition/micro-fabrication by photoinduced chemical reactions.

Response:

We thank the referee point the issue out. It is our fault that we didn't review these important and pioneering works in the manuscript. We added these two references (as reference [3] [10]) and rephrase the introduction in the revised manuscript. The revised content is shown below in the answer to the next question.

(2) In the introductory part, the authors claim that the complex process of lithography prevents the wide spread of quantum beam lithographic techniques. However, the lithographic nano/micro-fabrication techniques are now widely used in a lot of fields ranging from a laboratory level to mass production in factories. Difficulties have been resolved by the great effort of scientists and engineers although the process is actually complex. I suggest to the authors that they rephrase the point in a respectful form.

Response:

We apologize the inaccurate description in the introduction and remove the sentence “All these factors greatly limit the spread of the technology as well as in-field deployments” from the manuscript. Furthermore, we rewrite the whole introduction part. Now it reads:

“Conventional manufactory processes, ranging from circuit board printing down to integrated circuit and nano-devices fabrication, consist of multiple steps such as mask production, material deposition, photo-lithography and lift-off. Not only does each step add cost and chance to fail, it puts demanding requirements. For example, the sample surface temperature can be heated up too high during electron beam evaporation. Organic chemicals used in the photo-lithography process or high vacuum in deposition process can degrade the sample quality and put further limitations on the sample to use. Especially during lift-off, which is not trivial, faults lead to a waste of efforts and materials. And due to the complication of these procedures, advanced equipment and intensive training are required. On the other hand, precise positioning is a common difficulty under the current protocol, due to many factors, for example, the limited visibility of the sample surface covered with photoresists, moving samples among different setups in each procedure, imaging and aligning them without exposure. For the fabrications targeting on special sample locations, patterning while evaluating the sample quality is hard, as the process is not compatible with most characterisation measurements.

To invent a general patterned material deposition method with single step is desired to solve these problems. One of the key challenges is to find a mechanism to pattern the 2materials while depositing them in-situ. The mechanism has to be general enough that it can be applied to commonly used materials, and the deposition should be big and thick enough in scale, and have quality suitable for real applications. There are a variety of ways for patterning polymer, including thermal process, photochemistry and optical trapping [1-3]. However, to pattern large scale inorganic materials directly, especially metals is still challenging. In general, up to now, only photon based methods can reach high resolution and accuracy. But different methods are based on different light induced effects. The methods based on photothermal induced physical form changes usually requires high power lasers, nevertheless fabrication with high resolution and with materials like noble metals in small scales are difficult. Laser induced chemical reactions, such as polymerization and reduction, can deposit nanostructures with selected materials [4-6]. Optical trapping can also be used for depositing nanostructures either directly from their solution[7-9], or from photon-induced chemical reactions [10]. However, without a bonding mechanism among particles, the maximum size and thickness, the mechanical strength and electrical conductance are greatly compromised in the finished structures. There are also approaches by mixing metal particle with polymer and using polymer printing techniques to build metal composite structures. But the conductivity is usually orders of magnitude smaller than the bulk value. Moreover, for the polymer based methods [3, 6], after the deposition, the development of samples with organic solvents is required, this posts further limitations on both materials to be deposited and the substrate to use. To solve all these problems, in this work, we present a novel single-step direct material deposition method based on the combination of the photon-induced chemical reduction

and laser-induced optical trapping processes via semiconductor nanoparticles as the agent. The nanoparticles not only become a universal reduction agent and greatly broaden the selection of materials, but also can work as the seeds of growth trapped by the optical force as well as a "glue" to form rigid composite depositions. This new approach enables us to deposit a broad selection of materials with large area and thickness, high quality and high accuracy. It removes fabrication limitations posted by traditional methods and makes devices ready for practical applications."

(3) The description of Fig. 1a and 2b are missing and the illustration of Fig 1a left is misleading. The green spiral line in Fig. 1a shows circularly polarized beam? In the presence of light field, the "sensitizer nanoparticle" reduces metallic ions. But in Fig. 1a left, some of nanoparticles are already covered with reduced metal despite lack (or very small number) of photons outside the focus. The illustration in the middle of Fig. 1a gives little information (description) of the deposition mechanism. Give Fig. 1b words describing important stuffs, e.g., the contents of the two pipettes, green circular cone (this is maybe laser beam from the bottom), etc.

Figure 3 (Revised Figure 1 in the main text.) **Laser induced material deposition.** a, schematic illustrations of the principle of the LIMD method. (left) Schematic illustration of the material deposition process, i.e. under the optical field, the semiconductor nanoparticles are trapped towards the focus and on their surface photo-induced reduction reaction converts the metal ion into metal deposition. (middle) Ideal band structure of semiconductor particles. Photon excites electron from the valence band into the conduction band to become the free electron which triggers the photo-induced reduction reaction. (right) Schematic illustration of the photo-induced chemical reduction process. Free electrons excited by photon in the semiconductor particle (black sphere) reduce metal ions into metal particles (yellow sphere) on the surface of the semiconductor. b, schematic illustrations of the experimental procedure. (left) Drop-casted the thoroughly mixed reagents on the substrate surface, with one pipette contains metallate and the other contains semiconductor nanoparticles. (middle) a 532nm laser beam, focused by a microscope objective, creates deposition on the focal spot. (right) After the deposition and washing with water, the deposited pattern is left on the substrate. c, three samples of depositions on glass slides done with the LIMD method. The designs are shown in the upper row. The images in the lower row are the detail zoom-ins of the middle rows. (left) A logo of CUHK written in platinum. (centre) A yin-yang symbol written with platinum and gold. (right) A Chinese traditional ink painting written with platinum and gold. The white scale bars represent 50 μm , the blue ones represent 25 μm .

Response:

We are grateful to the reviewer point both problems out. We redraw the Fig. 1 a and b and add their descriptions in the revised manuscript.

(4) The 1st line of RESULTS, what are Part A and Part B?**Response:**

We are sorry for the missing description. The Part A is the metallic solution. Part B in the recipe works as the reducing agent for the chemical reduction process.

We rephrase the description at the beginning in the revised manuscript “*Part A (the metallate) and Part B (the semiconductor nanoparticles) are chosen based on the material to be deposited*” in the main text (Page 4, Paragraph 2). We also add these into Fig. 2.

(5) Discussion of the mechanism of the deposition is missing. In a related point, the stiffness of the deposition is also very important in the application of the method. The evaluation of the adsorption stiffness and mechanical strength improve the quality of the paper.

Response:

We thank the Reviewer for these two important suggestions.

We study the mechanism from several directions and summarize the results in Chapter 5 in the SI (Page 28-38). It includes the Model of deposition, Wavelength dependence, Laser power and chemical concentration dependence, and Analysis of the fine structures inside the deposition. Also related contents include the potential heating and optimization of the conductance are also included in the revised SI.

We also study the mechanical properties as suggested by the Reviewer in two directions. One is to measure the response in nanoindentation. The Young’s modulus measured in this experiment is similar to that of graphite. This is because the softer part in the composite, i.e. the ink particles, contribute to the deformation.

On the other hand, we performed multiple nanoindentations at the same point to evaluate the reversibility of the deformation as plotted in Fig. S24. Under 2 μN force (~ 13 GPa pressure) nanoindentation, the loading curves overlap with each other very well. The deposited structure was robust enough against such large indentation force, indicating that the deposited structures are mostly elastic under such indentation force.

The other measurement is to test the flexibility of the deposition. Normally, inorganic deposition is rigid, especially for metal depositions. Here, we find out the deposition can sustain a large rotation and the properties are still intact. This makes this deposition ideal for applications in flexible devices.

The details of the mechanical properties are written as Chapter 4.8 in the SI (Page 23-26).

(6) The photoexcitation by the laser produces an exciton in the nanoparticle and the excited electron can reduce a metal ion near the particle. At the same time, the photoexcited electron relaxes through a non-radiative pathway to generate heat. Especially, when carbon nanoparticles are used, I think the photothermal effect is not so small. The discussion on the photothermal effect is useful for readers who will use the method for heat-sensitive substrates/films of organic compounds.

Response:

We thank the Reviewer for this great point. This photothermal effect is indeed a factor to be considered. Since the reaction happens in the tiny vicinity of the semiconductor nanoparticles, a local temperature sensor with a high spatial resolution, close vicinity to the reaction centre and a good temperature resolution are needed. We take advantage of our expertise of using nitrogen vacancy (NV) centres in diamond as the sensor.

We spread nanodiamond particles on a glass substrate. We then use the LIMD method to deposit materials on nanodiamond surfaces and monitor the temperature change with NV centres *in-situ*. This is also a good demonstration that the LIMD method is compatible with optical measurements, thus, it can be used in parallel with them.

We found out that the maximum temperature rising due to the photothermal effect is around 10 Kelvin. This also explains why substrates like tape can be used with the LIMD method.

To be noted, this temperature change sensed by NV centres is a time-averaged effect. On the other hand, the laser excitation in this method is also a CW laser. Thus, a big temperature jump is not likely to happen. It is still worth in the future to use other methods to study this temperature effect, for example, by measuring with the transient absorption spectra with ultra-fast lasers (for example, Katayama *et al*, Langmuir 2014, 30, 9504–951).

We add a new section in SI (Chapter 6.2, Page 40-42) to explain the experiments and data in details.

(7) From the view point of "nano" fabrication, spatial accuracy of the material deposition method is crucial. It should be evaluated and discussed.

Response:

We thank the Reviewer for this helpful suggestion. We make several nanodots by depositing Platinum on glass slides as shown in Figure 5. The average size is around 400-500 nm.

There are three important parameters that limit the resolution of the LIMD. The size of the semiconductor particles, the penetration depth of the laser and the optical diffraction limitation. Currently, the smallest carbon nanoparticles we found is from the Kiwa-guro Ink, ultra-black. The average size of the particle (D in Figure 4) is ~90 nm obtained by the dynamic light scattering measurements (see SI Chapter 3.1, Page 5-7).

The penetration depth of laser (P in Figure 4) for the currently deposited materials is ranging from around 10 nm (conductors) to around 80 nm (insulators).

The optical diffraction limitation is roughly 200 nm-500 nm in our setup by using different objectives.

Currently we can deposit 400-500 nm Pt nanoparticles. This is close to the theoretical limit with the nanoparticles we use. Nevertheless, by choosing smaller nanoparticles, higher *N.A.* objectives, and optimizing recipe, we can reach better spatial resolutions with this method.

We group the SEM images of the nano-dots and single nano-wire together in Fig.3 (a,b) in the main text with added descriptions in the main text, and we add a new section in the SI (Chapter 4.9, Page 26-28) for the discussion of the spatial resolution.

Figure 4 Scheme of nanoparticle deposited by the LIMD method. The centre black sphere is the carbon particle, with diameter D . The yellow shell is the Pt deposited on the surface of the carbon particle, with thickness of P .

Figure 5 SEM images of smallest platinum dots obtainable so far with 2.68mw laser power. The yellow scale bar is 500 nm.

REVIEWERS' COMMENTS

Reviewer #1 (Remarks to the Author):

The authors explained very well the novelty in the updated version and corroborated it with several characterization. I do not have any further queries. The article now reaches the standards of Nat Comm and my review is to accept this article as is.

Reviewer #2 (Remarks to the Author):

In the report, the authors demonstrated micro-deposition of materials by using light induced material deposition (LIMD) method developed by them. The method seems versatile and is potentially applicable to many systems. For the previous version (1st submission) I concluded that the paper has the potential to be published in Nature Communications after appropriate revision and I pointed out concerns that should be addressed for improving the paper. In the revised version, the authors have sincerely responded to my questions and adequately addressed the concerns. I therefore can recommend the publication of the paper.